# MUC1: Structure, Function, and Clinic Application in Epithelial Cancers

**DOI:** 10.3390/ijms22126567

**Published:** 2021-06-18

**Authors:** Wenqing Chen, Zhu Zhang, Shiqing Zhang, Peili Zhu, Joshua Ka-Shun Ko, Ken Kin-Lam Yung

**Affiliations:** 1Division of Teaching and Research, School of Chinese Medicine, Hong Kong Baptist University, Hong Kong, China; 19481098@life.hkbu.edu.hk; 2Department of Biology, Hong Kong Baptist University, Kowloon Tong, Kowloon, Hong Kong, China; 15485412@life.hkbu.edu.hk (Z.Z.); 13480138@life.hkbu.edu.hk (S.Z.); pellyzhu@hkbu.edu.hk (P.Z.)

**Keywords:** MUC1, epithelial cancer, MUC1 cell barrier, tumor oncogene, therapeutic biomarkers, immunotherapy

## Abstract

The transmembrane glycoprotein mucin 1 (MUC1) is a mucin family member that has different functions in normal and cancer cells. Owing to its structural and biochemical properties, MUC1 can act as a lubricant, moisturizer, and physical barrier in normal cells. However, in cancer cells, MUC1 often undergoes aberrant glycosylation and overexpression. It is involved in cancer invasion, metastasis, angiogenesis, and apoptosis by virtue of its participation in intracellular signaling processes and the regulation of related biomolecules. This review introduces the biological structure and different roles of MUC1 in normal and cancer cells and the regulatory mechanisms governing these roles. It also evaluates current research progress and the clinical applications of MUC1 in cancer therapy based on its characteristics.

## 1. Introduction

Mucin1 (MUC1 also known as EMA, MCD, PEM, PUM, KL-6, and MAM6) is a macromolecular protein. It is the most readily recognized transmembrane protein of the mucin family and has a highly glycosylated extracellular domain. Under normal conditions, MUC1 covers the surface of all epithelial cells [1,2,3], forming a tight mesh. It forms a protective barrier through the mucosal surface and protects the cells from extreme environmental conditions [4,5,6,7,8,9]. In cancer cells, it has intracellular signaling functions and plays a significant role in cancer development.

MUC1 is commonly overexpressed in various epithelial adenocarcinomas such as lung, liver, colon, breast, pancreatic, and ovarian cancer. It is a well-known and vital oncogene that regulates various aspects of cancer (cell growth, proliferation, metastasis, apoptosis, developmental processes, etc.) by participating in different signaling pathways [6,10,11,12,13,14]. In the past, researchers have designed cancer therapeutic regimens based on MUC1 characteristics to maximize its potential as a therapeutic and prognostic target, used MUC1 as an immunogen to make vaccines, and controlled the expression of MUC1 to treat undesirable cancer-related phenomena. This review summarizes recent findings on the composition of MUC1, its functions in different tissues (healthy and diseased), and its clinical applications, with a special focus on the impact of MUC1 on various aspects of cancer.

## 2. The Structure of MUC1

The epithelium is a laterally connected single cell layer with apical-basal polarity. Due to its location, it is highly susceptible to external environmental influences and requires complex and well-developed defense mechanisms to maintain its integrity. Mucins are located in the epithelial layer, with secreted mucins having appeared early in the evolution of metazoans and formed transmembrane structures involved in the protection, repair, and survival of vertebrate epithelia [5,15,16,17]. The family of mucins to which MUC1 belongs is that of large, highly glycosylated proteins [7]. There are three types of mucins: trans-membrane (e.g., MUC1, MUC4, and MUC16), secreted (gel-forming) (e.g., MUC2, MUC5AC, and MUC6), and soluble (non-gel-forming) (e.g., MUC7, MUC8, MUC9, and MUC20) [17,18,19]. MUC1, the best-characterized transmembrane mucin, has a variable number of highly glycosylated, 20-amino acid tandem repeats (VNTR), a sperm protein-enterokinase-agarin (SEA) domain (extracellular), a transmembrane domain, and a 72-amino acid cytoplasmic tail domain that extends up to 200–500 nm out of the cell surface (Figure 1A) [20,21,22,23]. MUC1 is also thought to be encoded as a single polypeptide chain, which is then self-protectively cleaved immediately after translation at the GSVV motif (located within the SEA domain) into two peptide segments: the longer N-terminal subunit (MUC1-N) and the shorter C-terminal subunit (MUC1-C) [24,25,26,27]. These two subunits are linked through stable hydrogen bonds [1,28]. However, in cancer-related MUC1, its structure was found to have changed—overexpressed due to loss polarity in epithelial cells, and the carbohydrate side chain becomes uncomplete and forms new carbohydrate side chains (Thomsen-Friedenreich (TF or T), Tn, and sialyl-Tn (STn)) and exposure of the core peptide (Figure 1B) [3,29,30].

### 2.1. The N-Terminus

The MUC1 N-terminus, which is extracellular, contains highly conserved VNTR of 20 amino acids (HGVTSAPDTRPAPGSTAPPA), which are rich in proline (Pro or P), threonine (Thr or T), and serine (Ser or S) residues. The N-terminus are extensively modified by *O*-linked glycans (Ser and Thr residues) [5,8,29,31,32]. Notably, MUC1-N is the mucin component of the heterodimer that functions as a cell barrier, blocking cell-cell and cell-extracellular matrix interactions and protecting cells from cellular and pathogenic invasions while keeping the epithelium moist and repairing it [33]. In some cases, MUC1-N is released from the cell surface, leaving MUC1-C behind as a putative receptor that can be phosphorylated and involved in multiple signaling pathways associated with transformation and numerous signaling pathways related to tumor progression [1,7,34].

### 2.2. The C-Terminus

The C-terminus of MUC1 has been more extensively studied, especially in the context of inflammation and cancer development, due to its location. It includes the components mentioned above, a transmembrane structural domain of 28 amino acids, and a cytoplasmic tail of 72 amino acids [35]. The short and highly conserved cytoplasmic domain contains seven tyrosine residues and several serine and threonine residues, which represent potential docking sites for proteins with Src homology two domains and recognition sites for receptor tyrosine and other kinases. This includes ErbB receptors such as protein kinase C delta (PKCδ), glycogen synthase kinase 3β (GSK3β), and epidermal growth factor receptor (EGFR). As this domain is cytoplasmic, its function is more relevant in signal transduction [4,8,36].

## 3. Function of MUC1 in Normal Tissues

In healthy tissues, MUC1 functions as a barrier to protect cells mainly by virtue of its extracellular domain [37]. Membrane-bound MUC1 acts as a physical barrier through the action of its extracellular SEA domain, which can help regulate cell shedding and adhesion during metastasis; protect the apical cell membrane of epithelial cells from rupture, harmful environments, and immune attack; provide resistance to stimuli; inhibit immune responses through receptor shielding; and act as a decoy receptor for invading pathogens [7]. It is also involved in lubrication, cell surface hydration, and protection from degradative enzymes (Figure 2A) [18,38,39,40]. In the normal oral mucosal epithelium, MUC1, together with MUC5B and MUC7, exerts antimicrobial effects by continuously lubricating and stabilizing the mucus on the cell surface and conferring protection against proteolysis, thus preventing dehydration [1,41]. Sherry et al. showed that MUC1*, a transmembrane cleavage product of the MUC1 protein, can help propagate large numbers of pluripotent stem cells for therapeutic interventions [42].

## 4. Function of MUC1 in Cancer Tissues

In diseased tissues, the function of MUC1 has overly changed and tightly related to the treatment of and progression of different epithelial cancers (Figure 2B).

### 4.1. Pro- or Anti-Inflammatory Role

MUC1 can play either a pro- or anti-inflammatory role in different infection-induced cancers by acting as an immunomodulatory switch [37,43]. For example, in multiple sclerosis (MS), MUC1 plays an anti-inflammatory role and inhibits the dendritic cell (DC) response that is essential for inflammation to occur [44]. MUC1 can also affect Toll-like receptor (TLR) responses through immunomodulation. For example, paclitaxel can significantly alleviate cercal ligation and puncture (CLP)-induced acute lung injury in septic mice and in the lipopolysaccharide (LPS)-stimulated lung type II epithelial cell line A549 by activating MUC1 and inhibiting the TLR-4/nuclear factor (NF)-κB pathway [45].MUC1 can also inhibit TLR-4 expression by stabilizing hypoxia-inducible factor (HIF)-1α, thereby alleviating sepsis-induced lung injury and protecting organ function [46]. However, in its pro-inflammatory role, the tumor form of MUC1 can establish specific interactions with DCs and macrophages by controlling the recruitment of inflammatory cells, promoting tumor escape from the immune system, and creating a different inflammatory cell landscape in the tumor microenvironment [47]. When MUC1 acts in a pro-inflammatory manner, cancer cells use the vascular adhesion pathway of leukocytes in the inflammatory response to metastasis [48]. MUC1-C also induces epithelial-mesenchymal transition (EMT) by activating the inflammatory NF-κB p65 pathway, which activates the EMT transcriptional repressor zinc-finger E-box-binding homeobox 1 (ZEB1) [49]. Altered MUC1 glycosylation also promotes chronic inflammatory conditions that lead to malignant transformation and cancer progression [47]. It has been observed that MUC1 serves different inflammatory functions in different cancers. As an anti-inflammatory agent, MUC1 mainly affects DCs and thereby inhibits inflammation. As a pro-inflammatory agent, MUC1 promotes inflammation through different pathways by modulating related biomolecules in the tumor microenvironment and during EMT, and through altered glycosylation, contributing to the further progression to tumor formation.

### 4.2. Pharmacodynamic Inhibitors

MUC1 can lead to the emergence of drug resistance during cancer therapy as it is commonly overexpressed in various epithelial cancers. The overexpression of MUC1 has been reported to limit the effectiveness of fluorouracil (5-FU) by reducing intracellular drug uptake and anti-tumor drug action in pancreatic tumors [50]. MUC1 can also upregulate ATP-binding cassette (ABC)-B1 expression in an EGFR-dependent manner and induce chemoresistance [51]. MUC1-mediated nucleotide metabolism also plays a key role in promoting radiotherapy resistance in pancreatic cancer and can inhibit effective targeting through glycolysis [52]. MUC1 regulates the stabilization of HIF-1α and mediates the metabolic reprogramming of resistance to gemcitabine by increasing glucose uptake and thus enhancing resistance [53]. In hepatocellular carcinoma (HCC) cells, MUC1 can promote radioresistance by activating the Janus kinase (JAK)2/signal transducer and activator of transcription (STAT)-3 signaling pathway [54]. In addition, MUC1-C exerts oncogenic activity by targeting GalNAc-T5, a glycosyltransferase associated with tumor suppression in pancreatic cancer [55]. Clearly, drug resistance mediated by MUC1 occurs largely through the regulation of glycolytic metabolism, with some regulation of other specific pathways. A rise in glucose uptake can make tumor tissues less sensitive to increased drug concentrations; MUC1 amplifies this phenomenon, making it inextricably linked to drug resistance in tumors.

### 4.3. MUC1 Promotes Migration and Invasion of Cancers

MUC1 promotes the migration and invasion of a variety of cancers; the regulatory mechanisms governing this phenomenon are highly redundant and complex. The first mechanism is through the regulation of factors that are closely associated with tumor cell invasion and metastasis [56,57,58,59,60]. For example, EMT can be mediated through the transforming growth factor β (TGF-β) signaling pathway to promote the invasive and migratory capabilities of cancer cells. By treating HCC cells that express MUC1 with exogenous TGF-β1, TGF-β type I receptor (TβRI) inhibitors, TGF-β1 siRNA or activator protein 1 (AP-1) inhibitors, researchers have found that MUC1-induced autocrine TGF-β promotes cell migration and invasion through the c-Jun N-terminal kinase (JNK)/AP-1 pathway [60]. Silencing MUC1 expression also inhibits the migration and invasion of pancreatic cancer PANC-1 cells and induces apoptosis through the downregulation of the transcription factor Slug [61]. MUC1 also enhances the invasiveness of pancreatic cancer cells by inducing EMT [62]. The involvement of KL-6/MUC1 glycosylation in the metastasis of and invasion by pancreatic cancer cells has also been shown and may be related to the EMT process. Therapeutic strategies targeting KL-6/MUC1 glycosylation may therefore help to control the invasive behavior of pancreatic cancer cells [63]. MUC1-induced invasion and proliferation have also been shown to occur through the increased production of exogenous platelet-derived growth factor (PDGF)-A [64].

In rectal cancer, the sialo-oligosaccharide form of MUC1 has been shown to determine the metastatic potential of colorectal cancer cells and to have clinicopathological utility in evaluating the outcome and prognosis of patients [13]. The crosstalk between MUC1 and c-Met in HCC patients could be advantageous for HCC cell invasion through modulation of the β-catenin/c-Myc pathway. Therefore, MUC1 and c-Met are potential therapeutic targets for HCC [65].

Cancer cells survive hypoxic environments by promoting the expression of pro-angiogenic genes to stimulate angiogenesis [24]. The expression of MUC1 promotes angiogenesis in cancer and, to a certain extent, promotes tumor migration and invasion. MUC1 and vascular endothelial growth factor (VEGF) expression in human breast cancer are highly correlated, and it has been demonstrated that MUC1 expression promotes angiogenesis in human breast cancer both in vivo and in vitro [66,67]. MUC1 induces angiogenesis in tumor microenvironments by increasing the expression of neuropilin-1 (NRP1, a co-receptor of VEGF) and its ligand VEGF. Alternatively, MUC1 can activate intracellular signaling pathways such as Ras/mitogen-activated protein kinase (MAPK), JAK/STAT, and phosphoinositide 3-kinase (PI3K)/Akt/mammalian target of rapamycin (mTOR) to increase the expression of VEGF [62,66,68].

### 4.4. MUC1 Inhibits Cancer Cell Growth and Apoptosis

MUC1 is also involved in the regulation of different pathways of cancer cell growth and apoptosis. Transformation and the loss of polarity in breast cancer epithelial cells causes the overexpression of MUC1, which in turn induces Crumbs homolog-3 (CRB3) expression and inhibits yes-associated protein (YAP) and YAP/β-linked protein-mediated Myc expression. The overexpression of MUC1-C, but not MUC1-N, is also sufficient to induce transformation and resistance to stress-induced apoptosis [15,69]. MUC1 contributes to the growth and survival of pancreatic cancer cells by activating the MAPK pathway; pharmacological inhibition of this pathway inhibits the proliferation of MUC1-expressing cells [70]. The knockdown of MUC1 has been shown to inhibit cell proliferation, enhance cell-cell aggregation, and induce apoptosis [71]. MUC1 also activates JNK1 and inhibits cisplatin-induced apoptosis in human colon cancer HCT116 cells. The pharmacological inhibition or knockdown of JNK significantly reduces the ability of MUC1 to inhibit cisplatin-induced apoptosis in response to genotoxic anticancer drugs [72]. Cell permeability inhibitors such as protein transduction domain MUC1 inhibitory peptide (PMIP), which interfere with MUC1-EGFR interactions, have been extensively developed and shown to effectively kill breast cancer cells both in vitro and in tumor models [73].

Overall, MUC1 affects a variety of tumor progression pathways. Pancreatic CD133^+^ cells exhibit higher expression levels of MUC1, contributing to their tumorigenic phenotype through increased interactions between MUC1-C and β-catenin, which in turn modulate oncogenic signaling cascades [31]. In HCC cells, MUC1 overexpression promotes HCC progression and tumorigenesis through the JNK/TGF-β signaling pathway [74]. In hematological malignancies, MUC1-C has been associated with various pathways related to disease pathogenesis, such as Wnt/β-catenin, fms-like tyrosine kinase 3 (FLT3), breakpoint cluster region protein (BCR)/Tyrosine-protein kinase (ABL), and NF-*κ*B, which are involved in tumorigenesis [26]. The aberrant expression of MUC1-C is also sufficient to induce transformation and block cell death in response to genotoxic, oxidative, and hypoxic stresses [75]. In epithelial ovarian cancer tissues, the overexpression of tumor-associated MUC1 and its multiple biological functions contribute to cell-cell adhesion, signaling, migration, proliferation, and differentiation in cancer cells; the regulation of MUC1 in malignant cells may therefore alter these carcinogenesis pathways [76].

## 5. Clinic Application

When it comes to the clinic application of MUC1, researchers are more focused on its utilization in therapeutic clinical diagnosis and immunotherapy (Figure 3).

### 5.1. Therapeutic Marker

MUC1 is frequently used as a therapeutic marker in clinical applications because of its aberrant overexpression in various epithelial cells. Its receptor-like extracellular domain can be released into the external environment and act as a decoy for mucosal pathogens, sensing the external environment. The protein can also activate intracellular signaling pathways through its cytoplasmic structural domain [8].

MUC1 is also a well-established and useful biomarker for the early detection of gastric cancer (GC) [77]. The expression levels of MUC1 and MUC5AC correlate significantly with the tumor grade of colorectal cancer and are therefore used as markers for assessing the prognosis of GC patients [78,79]. In pancreatic cancer, MUC1 can be detected in different stages and can indicate initiation and progression with good diagnostic accuracy [80]. MUC1 can also be used as a biomarker for pigeon-sensitive asthma patients and not just a negative predictor of the survival of patients with cancers of epithelial origin [81]. In addition, MUC1 has been associated with immune checkpoint genes, neoantigens, and certain prognostic indicators of immunotherapy such as tumor mutational burden (TMB) and microsatellite instability (MSI), suggesting that it may also serve as a target and prognostic biomarker for immunotherapy.

To further improve the detection accuracy and sensitivity of MUC1 biomarkers, some groups have designed sensors with different novel materials to detect and quantify MUC1 with higher efficiency. For example, MUC1 inducer-linked PtAu nanoparticles (NPs) were developed from nanomaterials with unique physical and chemical properties to improve the selectivity and sensitivity of the colorimetric detection of dual cancer markers; this has aided the explicit recognition of the MUC1 proteins on the surface of cancer cells [82]. Directly competitive electrochemical immunosensors based on gelatin modifications of dopamine (DA)/MUC1-functionalized electroactive carbon nanotubes have also been designed to be used as signal generation probes for the early diagnosis of breast cancer. The gelatin-modified electrodes were used as supports to immobilize antibodies (anti-MUC-1), and the electrochemical reactions of functionalized electroactive carbon nanoprobes were used for the quantitative measurement of MUC-1 [83]. New radiolabeled conjugates and peptides with desirable biological properties have also been used to target tumor-specific MUC1 antigens to diagnose and treat cancers [84]. Optical and electrochemical platform-based ensemble biosensors and nanosensors for detecting and quantifying MUC1 have also been developed with electrochemiluminescence sensors to detect MUC1 protein exocytosis in breast cancer cells and MUC1 protein in their derived exosomes [85,86].

### 5.2. Immunotherapy

MUC1 has also made a notable contribution to immunotherapy. For immunotherapy, MUC1 research has followed three main lines [87]. The first is the glycopeptide epitope of MUC1, whose role in the induction of humoral and cellular adaptive immune responses has been recognized for many years [88]. The second is MUC1 expressed by cancer cells can affect the phenotype and function of immune cells in the tumor microenvironment. Last, focusing on smaller membrane-spanning component (MUC1-C) has increased. According to the three lines, MUC1 was designed to be an antibody-based therapy, immune cells hijacking therapy, and vaccine-based therapy.

First is the glycopeptide epitope of MUC1. The antibody PankoMab-GEX, a glycopeptide epitope located in the TR structural domain, can react with conformational epitopes in which the threonine in PDTRP carries Tn or T glycans and selectively reacts with cancer mucins [89]. PankoMab-GEX has been humanized and glycan-optimized to enhance ADCC and ADCP activity and enhance NK cell killing [87]. Meanwhile, both MUC1-N (hyperglycosylated) and MUC1-C components contribute to immune evasion of cancer cells, which may also be an essential consideration in developing MUC1-based immunotherapy strategies [87]. In addition, hyperglycosylated MUC1 can inhibit its processing and presentation to T cells by DCs as tumor antigens, thus blocking anti-tumor immune responses. Vaccines targeting oligosaccharylated MUC1 are being developed to induce antibodies and T cells to eliminate inflammatory and/or tumor-initiating cells expressing this form of MUC1, thereby preventing further inflammation promoting the anti-tumor activity of many effector cells in the microenvironment [47].

In recent years, attention has increasingly turned to the role of MUC1-C in immunotherapy and can affect immune cells in the tumor environment. For example, in lung cancer, Lewis lung cancer cells expressing MUC1-C (LLC/MUC1) exhibit upregulation of Programmed death-ligand 1 (PD-L1) and inhibition of interferon-γ (IFN-γ). In addition, MUC1-CPD-L1 signalling promoted activation of CD8C T cells, demonstrating that MUC1-C is a potential target for reprogramming the tumor microenvironment [90]. Antibodies specific for the extracellular structure of MUC1-C have also been isolated and screened from them to inhibit the invasion of triple-negative breast cancer [91]. The reason is that genetic or pharmacological targeting of the oncogenic MUC1 subunit MUC1-C enhances the transcription rate of the immune checkpoint ligand PD-L1 in triple-negative breast cancer (TNBC) cells by recruiting MYC and NF-kB p65, which in turn inhibits PD-L1 expression [92]. In addition, MUC1-C has emerged as an attractive target for the development of mAb-based therapeutics. For example, mAb 3D1 is an antibody against the non-shed MUC1 C-terminal subunit that binds to the restricted α3 helix of the extracellular structural domain of MUC1-C with a low nM affinity. Its reactivity is selective for human cancer cell lines and primary cancer cells that express MUC1-C. It also binds to monomethyl whey protein E (MMAE) to form mAb 3D1-MMAE antibody-drug conjugate (ADC), which kills MUC1-C positive cells in vitro while being non-toxic to MUC1 transgenic (MUC1.Tg) mice and active against human HCC827 lung tumors [93]. The concomitant use of anti-MUC1-C/NPs antibodies prolonged their retention in the tumor microenvironment in vivo and ensured that the radio-enhancing effect of NPs was maintained. By concomitant administration with radiotherapy (XRT), the efficiency of XRT was significantly improved, significantly enhancing the inhibition of tumor growth and prolonging the overall survival of the animals [94]. Several researchers have also isolated specific antibodies against the extracellular structure of MUC1-C that recognize recombinant MUC1 and the native MUC1-C protein in breast cancer cells and screened for antibodies that highly inhibit the invasion of triple-negative breast cancers. This could be a very effective therapeutic candidate for human breast cancer, especially triple-negative breast cancer (TNBC) [91,95]. There are also monoclonal antibodies specific for the extracellular region of the MUC1 subunit MUC1-C (anti-hMUC1 antibody and antibody GP1.4), the former recognizes the MUC1-C protein in pancreatic cancer cells, thereby inhibiting epidermal growth factor (EGF)-mediated extracellular signal-regulated kinase (ERK) phosphorylation and cell cycle protein D1 expression and suppressing MUC1 in vitro and in vivo functions of MUC1. The latter triggered the internalization of EGFR in pancreatic cancer cells, leading to the inhibition of EGF-stimulated ERK phosphorylation, thus inhibiting the proliferation and migration of pancreatic cancer cells. This is a promising targeted therapy that could be further developed to treat pancreatic cancer [96,97].

MUC1 currently ranks second among 75 candidate antigens for cancer vaccines and over the years, has been used as the basis for the development of different types of vaccines [98]. The first is a vaccine that uses MUC1 alongside tumor-associated carbohydrate antigens (TACAs), such as Tn and STn, with variable number tandem repeats (VNTRs) as immunogens. However, its therapeutic efficacy is not yet sufficiently high for clinical use. This may be due to the low immune tolerance and xenobiotic nature of TACA-MUC1 [99]. Some groups have used different antigenic carrier proteins such as bovine serum albumin or keyhole bug hemocyanin conjugated to MUC1 [100], whereas others have designed MUC1 glycopeptide mimics in which the galactose-galactosamine disaccharide is linked to threonine (TF antigen) via an unnatural β-glycosyl bond. The resulting MUC1-β-TF is more stable against glycosidases that can cleave this sugar from the corresponding MUC1 glycopeptide with a natural α-TF linkage [101].

In the emerging field of targeted delivery, different antibodies or inducers against MUC1 have proven helpful for tracking cancer cells [98]. Various adjuvants have been used alongside MUC1 glycopeptides to enhance their immunogenicity. For example, fully synthetic multicomponent vaccines have been synthesized by incorporating different T helper cell epitopes and TLR agonists [100]. Alternatively, vaccine candidates with MUC1 glycopeptide epitopes and the lipopeptide adjuvant Pam2Cys have been shown to elicit MUC1-specific antibodies and cytotoxic T lymphocyte (CTL) responses in the absence of other injected lipids or adjuvants [102]. NPs have been developed to deliver mRNA vaccines that encode the tumor antigen MUC1 to DCs in lymph nodes, thereby facilitating the activation and expansion of tumor-specific T cells [103]. The construction of MUC1 glycopeptide vaccines by presenting α-GalCer adjuvants and antigens on gold NPs has also been proposed; this technique can potentially enhance anti-tumor responses during cancer immunotherapy [104]. The MUC1 vaccine has been tested in combination with other drugs to increase its anti-tumor effects. Researchers have combined a vaccine consisting of the MUC1 core peptide with indomethacin to stimulate tumor-specific immune responses and alleviate the immunosuppressive microenvironment within breast tumors. Compared to the administration of the vaccine alone, its combination with the drug reduced tumor cell proliferation and increased tumor cell apoptosis, making the cells susceptible to killing by immune cells [105]. Vaccines with the immune adjuvant Pam3CysSK4, the peptide T-helper epitope, and the aberrantly glycosylated MUC1 peptide have also been designed. The covalent linkage of these three components is essential for maximum efficacy. This vaccine produces a CTL that recognizes both glycosylated and non-glycosylated peptides, thereby effectively circumventing the failure of CTLs and IgG antibodies in targeting cancer expressing MUC1 due to conformational dissimilarities. The three-part vaccine can also overcome the antigen-processing sensitivity of the densely glycosylated MUC1 peptide, unlike similar non-glycosylated vaccines that produce a CTL that recognizes only non-glycosylated peptides [106].

MUC1 DNA vaccines have also been investigated. pcDNA3.1-VNTR is a MUC1 DNA vaccine composed of VNTRs that induce a significant MUC1-specific CTL response; it has preventive and therapeutic effects against pancreatic cell Panc02 -MUC1 tumors [107]. In mice, the immune response to MUC1 DNA vaccination was seen to effectively suppress CD4+ T cells in colon cancer cells transfected with MUC1 cDNA [108].

## 6. Conclusions

MUC1, a highly glycosylated protein, has two subunits that play different roles in normal and diseased states. In normal conditions, the N-terminus serves as a barrier to protect cells from harmful environmental and bacterial infections. In tumors, the C-terminus is localized inside cells and is more involved in different signaling pathways that influence and regulate tumor growth, survival, invasion, migration, and apoptosis. MUC1 is undeniably an oncogene that transforms inflammation into cancer, enhances drug resistance, promotes tumor metastasis, and has been shown to play a significant role in cancer progression. Based on these characteristics, researchers have designed different MUC1-targeting therapeutic approaches to treat cancer. Unlike in other organs, the overexpression of MUC1 has been found in many types of epithelial cancers, making it a good marker for diagnosis and prognosis in clinical treatment.

In the emerging field of tracking the targeted delivery of therapeutic agents to cancer cells, researchers have designed different types of vaccines aimed against the overexpression of MUC1, which can be effective in cancer prevention and treatment. The development of MUC1-based immunotherapeutic strategies has also been reported. This review has highlighted the structural features of MUC1 and its primary functions in cancer progression. Its clinical applications developed over the last decade were also discussed.

Although there have been many contributions to the study of MUC1, its role in different aspects of cancer progression, and its utility as a biomarker, it is only after scientists harnessed materials science to design detectors that the precise and quantitative determination of MUC1 in cancers was made possible. Despite these advances, the detailed mechanisms by which MUC1 exerts its effects remain to be explored. Even though MUC1 vaccines and antibodies are being developed, with some having shown significant therapeutic effects and currently undergoing phase I/II clinical trials, certain side-effects and clinical safety issues need to be further explored and confirmed [3,109,110]. In conclusion, MUC1 is inextricably linked to various epithelial cancers, mandating further explorations of this phenomenon.

## Figures and Tables

**Figure 1 ijms-22-06567-f001:**
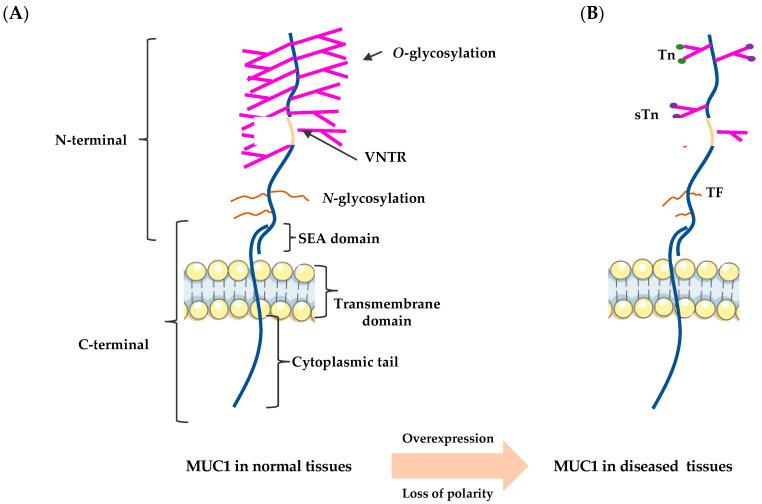
Structure of MUC1 in normal tissues and diseased tissues. (**A**) The structure of MUC1 in normal tissues; (**B**) The structure of MUC1 in diseased tissues.

**Figure 2 ijms-22-06567-f002:**
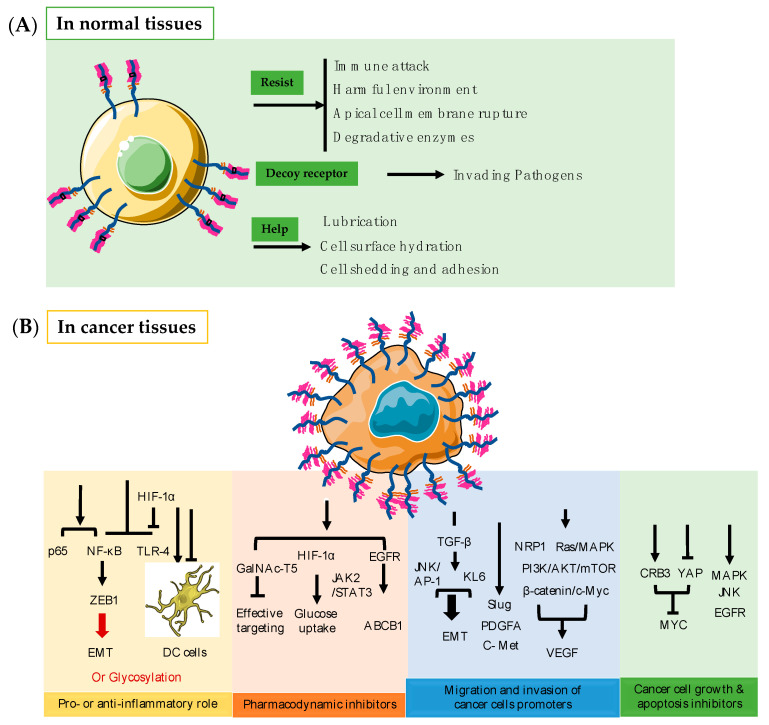
Different function of MUC1 in health or cancer tissues. (**A**) The function of MUC1 in normal tissues; (**B**) The function and its main pathways of MUC1 in cancer tissues.

**Figure 3 ijms-22-06567-f003:**
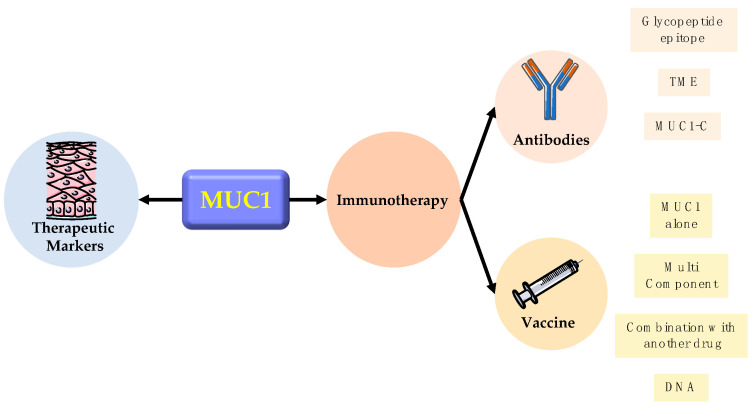
Clinical applications of MUC1.

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
