# Peer review of "MUC1: Structure, Function, and Clinic Application in Epithelial Cancers"

_ijms, 2021, doi:10.3390/ijms22126567_

Round 1

Reviewer 1 Report

International Journal Molecular Sciences

ijms-1232504

“MUC1: Structure, function, and clinic application in epithelial cancers”

Submitted by Joshua Ka-Shun Ko and Ken Kin-Lam Yung

The work submitted by the authors makes a revision of mucin-1 (MUC1) glycoprotein highlighting its role in health and in cancer as well as its application in diagnostic and in immunotherapy. Due to the relevance of this glycoprotein there are several recent reviews in the literature that already extensively describe MUC1 structure, function, and its application in the development of strategies targeting cancer. Some of recent literature about MUC1 can be found in the following manuscripts: Chem Soc Rev 2017 Nov 27;46(23):7154-7175. doi: 10.1039/c6cs00858e; Semin Immunol. 2020 Feb;47:101389. doi: 10.1016/j.smim.2020.101389; IMMUNOTHERAPY VOL. 12, NO. 17, doi.org/10.2217/imt-2020-0019 Biomed Pharmacother. 2020 Dec;132:110888. doi: 10.1016/j.biopha.2020.110888; Front. Cell. Infect. Microbiol., 24 April 2019 | https://doi.org/10.3389/fcimb.2019.00117, among others. For that reason, the actual manuscript lacks novelty if we compare with the preexisting literature. The authors also claim structure in the title, but they did not include the tumor-associated glycan structures and the peptide sequence of the tandem repeated domain of N-terminal of MUC1. Figure 1 is not properly prepared the schematic representation of the glycan antigen should be reconsidered since it is not clear the difference between the antigens. Therefore, unfortunately I regret to inform that I do not support this revision work for publication in IJMS.     

Author Response

First of all, thank you to patiently reviewourpaperand your detailed comments. And the followingareourresponsesto your comments.Point1Due to the relevance of this glycoprotein there are several recent reviews in the literature that already extensively describe MUC1 structure, function, and its application in the development of strategies targeting cancer. Some of recent literature about MUC1 can be found in the following manuscripts: Chem Soc Rev 2017 Nov 27;46(23):7154-7175. doi: 10.1039/c6cs00858e; Semin Immunol. 2020 Feb; 47:101389. doi: 10.1016/j.smim.2020.101389; IMMUNOTHERAPY VOL. 12, NO. 17, doi.org/10.2217/imt-2020-0019 Biomed Pharmacother. 2020 Dec; 132:110888. doi: 10.1016/j.biopha.2020.110888; Front. Cell. Infect. Microbiol., 24 April 2019 | https://doi.org/10.3389/fcimb.2019.00117, among others. For that reason, the actual manuscript lacks novelty if we compare with the preexisting literature.Response1:i.Chem Soc Rev 2017 Nov 27;46(23):7154-7175. doi: 10.1039/c6cs00858eThis reviewis different from our review, though it also described the structure of MUC1 and the MUC1 utilization in designing vaccines. This review is more focusedon the actualmolecular structure of MUC1, its derivatives,and theirchangesin solid or solutionin the molecular filed. And for our review, we focus on the MUC1, its potential and reality contribution in clinical cancer and other diseasetreatment, including immunotherapy and therapeutic markers.These two reviews actually described the MUC1 in different prospect.ii.Semin Immunol. 2020 Feb;47:101389. doi: 10.1016/j.smim.2020.101389This review well described the structure of MUC1, it detailly shows the MUC1 structure in normal and cancer cells, especially for glycan chains composited by kinds of monosaccharides. Compared to this review, our review additionally conclusion the regulation mechanism of MUC1 in cancer invasion, metastasis, angiogenesis, and apoptosis and list the signaling pathway by figure.iii.IMMUNOTHERAPY VOL. 12, NO. 17, doi.org/10.2217/imt-2020-0019This review mainly concentrated on the immunotherapy of MUC1 about antibody-based therapeutics. And our review is more willing to describe MUC1 in many different fields including its applications in immunotherapy. Undeniably this review mentioned more anti-MUC1 antibodies, which we have cited and added in our review.iv.Biomed Pharmacother. 2020 Dec;132:110888. doi: 10.1016/j.biopha.2020.110888Similar to Paper2, it described the structure of MUC1, and MUC1-based cancer vaccines. Our review not only focus on its structure and utilization in immunotherapybut also its mechanism in the epithelial cancer invasion, metastasis, angiogenesis, and apoptosis.v.Front. Cell. Infect. Microbiol., 24 April 2019 | https://doi.org/10.3389/fcimb.2019.00117This paper highlights the structure of MUC1, and one of its function-barrier, also the role of inflammation. Our review also mentioned the three topics, but our review also contains other information about MUC1.

In terms of our review as whole, we comprehensively described the MUC1 in many aspects (structure, function and clinic application). These papers most focus on MUC1 structure, utilization of immunotherapybut all of them describingMUC1 from different aspects compare to our review.Therefore, these papers exactly can be cited in our review to enrichingour review.Point2The authors also claim structure in the title, but they did not include the tumor-associated glycan structuresResponse 2:Actually, our review mentioned it at the end of title 2-the structure of MUC1.The introduction of MUC1 structure on cancer cellsis not very complex, so we put it together with the normal MUC1 structure and introduced it.We explained in cancer cells, thecause of carbohydrate side chainschangesand its differences from the normal one. Point3the peptide sequence of the tandem repeated domain of N-terminal of MUC1”Response 3:Inthe structure of MUC1 part, we mentioned the tandem repeat domain of N-terminal of MUC1(VNTR), a variable number of highly glycosylated, 20-amino acid tandem repeats (VNTR)and we will revise itsabbreviationsand its sequenceto make it more straightforward.Point4Figure 1 is not properly prepared the schematic representation of the glycan antigen should be reconsidered since it is not clear the difference between the antigens.Response 4:InFigure1, in normal cells, MUC1 is highlyglycosylation and one is O-glycosylationantigens, the other is N-glycosylationantigensshows in different shapes,which we have revised and pointin Figure1.In cancer cells, the glycans havechanged, sTn, Tn, TF their difference hasalso presentedand been revisedinFigure1.And we have further described it in the text.The glycan antigens are composed of kinds of monosaccharides. However, since it is not as closely related to the full paper and is lengthier, we decided not to elaborate on the composition of glycosylated antigens(the monosaccharide part) in detail but to show the differences in a more macroscopic way in the figure.At last, we sincerely thankyou for your comments which helpus to improve and enrich our review.

Reviewer 2 Report

This review well described the function of MUC1 in normal and cancer cells. In addition, the clinical application of MUC1 has been well described. If information on the development of therapeutic antibodies targeting MUC1 is added in the immunotherapy section based on the following references, a lot of information is likely to be provided to researchers.

  1. Panchamoorthy, G., Jin, C., Raina, D., Bharti, A., Yamamoto, M., Adeebge, D., Zhao, Q., Bronson, R., Jiang, S., Li, L., Suzuki, Y., Tagde, A., Ghoroghchian, P. P., Wong, K. K., Kharbanda, S., & Kufe, D. (2018). Targeting the human MUC1-C oncoprotein with an antibody-drug conjugate. JCI insight, 3(12), e99880.
  2. Detappe A, Mathieu C, Jin C, Agius MP, Diringer MC, Tran VL, Pivot X, Lux F, Tillement O, Kufe D, Ghoroghchian PP. Anti-MUC1-C Antibody-Conjugated Nanoparticles Potentiate the Efficacy of Fractionated Radiation Therapy. Int J Radiat Oncol Biol Phys. 2020 Dec 1;108(5):1380-1389.
  3. Kim MJ, Choi JR, Tae N, Wi TM, Kim KM, Kim DH, Lee ES. Novel Antibodies Targeting MUC1-C Showed Anti-Metastasis and Growth-Inhibitory Effects on Human Breast Cancer Cells. Int J Mol Sci. 2020 May 5;21(9):3258.
  4. Wu G, Maharjan S, Kim D, Kim JN, Park BK, Koh H, Moon K, Lee Y, Kwon HJ. A Novel Monoclonal Antibody Targets Mucin1 and Attenuates Growth in Pancreatic Cancer Model. Int J Mol Sci. 2018 Jul 9;19(7):2004.
  5. Wu G, Kim D, Kim JN, Park S, Maharjan S, Koh H, Moon K, Lee Y, Kwon HJ. A Mucin1 C-terminal Subunit-directed Monoclonal Antibody Targets Overexpressed Mucin1 in Breast Cancer. Theranostics. 2018 Jan 1;8(1):78-91.
  6. Hisatsune A, Nakayama H, Kawasaki M, Horie I, Miyata T, Isohama Y, Kim KC, Katsuki H. Anti-MUC1 antibody inhibits EGF receptor signaling in cancer cells. Biochem Biophys Res Commun. 2011 Feb 18;405(3):377-81.

Author Response

If information on the development of therapeutic antibodies targeting MUC1 is added in the immunotherapy section based on the following references, a lot of information is likely to be provided to researchers.Response1: First of all,thanks for your review comments, and “thedevelopment of therapeutic antibodies targeting MUC1”are primarilyin targeting MUC1-C field. Wefinally decided to add this pointto “Title5.2 immunotherapy” the end of the MUC-C part andcited the papers you mentioned above.
